# Preparation and Primary Bioactivity Evaluation of Novel Water-Soluble Curcumin-Loaded Polymeric Micelles Fabricated with Chitooligosaccharides and Pluronic F-68

**DOI:** 10.3390/pharmaceutics15102497

**Published:** 2023-10-20

**Authors:** Pattarachat Ingrungruengluet, Dingfu Wang, Xin Li, Cheng Yang, Yaowapha Waiprib, Chunxia Li

**Affiliations:** 1Shandong Key Laboratory of Glycoscience and Glycotechnology, Key Laboratory of Marine Drugs of Ministry of Education, School of Medicine and Pharmacy, Ocean University of China, Qingdao 266003, China18263049928@163.com (X.L.); acheng0912@163.com (C.Y.); 2Department of Fishery Products, Faculty of Fisheries, Kasetsart University, Bangkok 10900, Thailand; 3Center for Advanced Studies for Agriculture and Food (CASAF), Kasetsart University Institute for Advanced Studies, Kasetsart University, Bangkok 10900, Thailand

**Keywords:** curcumin, chitooligosaccharides, pluronic F-68, polymeric micelles

## Abstract

Curcumin (CU) is a bioactive compound extracted from turmeric and has various advantages. However, the benefit of CU is limited by its low water solubility (11 ng/mL). This research aimed to fabricate a water-soluble CU nano-formulation with chitooligosaccharides (COS) and pluronic F-68 (PF) utilizing the polymeric micelle method. The optimized curcumin-loaded chitooligosaccharides/pluronic F-68 micelles (COSPFCU) exhibited high encapsulation efficiency and loading capacity (75.57 ± 2.35% and 10.32 ± 0.59%, respectively). The hydrodynamic diameter of lyophilized COSPFCU was 73.89 ± 11.69 nm with a polydispersity index below 0.3. The COSPFCU could be completely redispersed in water and showed high DPPH scavenging ability. Meanwhile, COSPFCU could significantly reduce the cytotoxicity of the RAW 264.7 cells compared to native CU. Furthermore, COSPFCU improved the inhibition of NO release activity at 72.83 ± 2.37% but 33.20 ± 3.41% for the CU, with a low cytotoxicity concentration in the RAW 264.7 cells.

## 1. Introduction

Curcumin (CU) is classified as a member of the curcuminoid group, which comprises polyphenolic chemicals [1]. It is derived from turmeric (*Curcuma longa*) in Zingiberaceae with yellow-orange color [1,2]. Its potential bioactivity has been thoroughly examined, and includes antimicrobial [3,4,5], antifungal [6,7], antioxidant [8,9,10], anti-inflammatory [9,11,12], and anti-cancer [13,14] bioactivities. CU has been extensively employed as a food supplement, coloring agent, preservative, and traditional medicine. However, the advantage of CU is limited by its low water solubility (11 ng/mL) and low absorption rate [15]. Considering its inherent instability, reactivity, and restricted bioavailability, some researchers even made the case that performing more clinical trials on curcumin may be unnecessary [16]. To address this issue, several techniques have been explored to improve the effectiveness, bioavailability, and solubility of such polyphenol compounds. These include employing nano-formulation techniques such as precipitation, homogenization, microemulsions, solid-dispersion, nanoparticles, and polymeric micelles [17].

Polymeric micelles serve as a drug delivery system fabricated from hydrophobic molecules (core), co-polymers (core-shell), and hydrophilic bioactive molecules (outer-shell), respectively [18]. This encapsulation technique has gained significant attention due to its advantageous properties, such as high drug-loading capacity, targeted delivery, long circulation, ease of production, and controlled release through thermosensitive mechanisms [19,20,21,22,23]. Pluronic, also known as a poloxamer, is a widely utilized nonionic co-polymer for polymeric micelle formation. This polymer is a triblock co-polymer composed of hydrophilic poly(ethylene oxide; PEO) and hydrophobic poly(propylene oxide; PPO), organized in a PEO-PPO-PEO block configuration [24]. The pluronic exhibits self-assembly in an aqueous solution, resulting in the formation of micelles which consist of a PPO core surrounded by a PEO corona, depending on the ratio of PPO and PEO in the formula [25]. Pluronic F-68 (PF) has a molecular weight of 8.4 kDa, a hydrophilic/lipophilic balance (HLB) of >24, and 80% PEO [26]. PF is highly soluble in water and dispersible across a broad spectrum of solvents and temperatures, enabling its utilization across a diverse array of applications [27]. However, using polymeric micelles with the sole co-polymer as prolonged drug carriers is not feasible in practice due to the tendency of particle aggregation and physical deterioration under different concentrations and temperatures [28]. The outer shell of polymeric micelles performs a pivotal function in enhancing the bioactivity and characteristics of the particles.

Chitosan and its derivatives (β-(1–4)-poly-N-acetyl-D-glucosamine) are marine polymers that are extracted from the hard exoskeleton of subphylum crustacea using the chemical method [29]. When the molecular weight (Mw) and degree of polymerization (DP) of chitosan are below 3.9 kDa and 20, respectively, that is called chitooligosaccharides (COS) [30]. COS possess low viscosity and solubility in aqueous solutions due to the greater number of functional groups accessible for reaction with water molecules [29]. Due to its biological activities, which include anti-inflammatory, non-cytotoxic, biodegradable, and other properties, COS has a wide variety of potential uses in the domains of pharmacology, medicine, industry, and agriculture [29,30,31,32]. For polymeric micelle formulations to improve their ability to transport drugs, COS was chosen as the outer-shell material for a number of reasons. For instance, by conjugating with the amide groups in the structures, COS can increase the hydrophobic compounds’ water solubility [33,34].

The objectives of this study were to fabricate water-soluble polymeric micelles to load the hydrophobic molecules of CU, utilizing COS (outer-shell) and PF (core-shell) as the materials. The CU-loaded polymeric micelles were evaluated for encapsulation efficiency and loading capacity to indicate the potential of the system. The characterization of polymeric micelles was studied, including dynamic light scattering analysis, transmission electron microscopy, X-ray diffraction, in vitro release profile, and DPPH scavenging activity. The cytotoxicity of polymeric micelles and NO release inhibition in the RAW 264.7 cells was evaluated (Figure 1).

## 2. Materials and Methods

### 2.1. Materials

Chitooligosaccharides (COS) (2022052501CWR) (Mw 340–1610 Da, >90% degree of deacetylation) was obtained from Qingdao Hehai Biotech Co., Ltd., Qingdao, China. Curcumin (CU) (114G022) was purchased from Solarbio^®^, Beijing, China. Pluronic F-68 (PF) (TC222) was obtained from Himedia Laboratories Pvt. Ltd., Thane, India. Ascorbic acid (KA79) was purchased from Kemaus, Australia. Phosphate buffered saline pH 7.4 was purchased from Sinopharm Chemical Reagent Co., Ltd., Shanghai, China. DPPH (2,2-Diphenyl-1-picrylhydrazyl) (D9132) was purchased from Sigma-Aldrich, Roedermark, Germany. Ethanol (E7025-1-2501, 99.99%) was obtained from QRëC, Auckland, New Zealand. All these chemicals were analysis grade.

### 2.2. Optimization of Conditions of COSPFCU Fabrication

The COSPFCU polymeric micelles were prepared according to previous reports with slight modifications [35]. The effect of PF concentration and fabrication temperature was investigated.

The PFCU micelle was synthesized by introducing a 0.1% *w*/*v* CU solution (in ethanol) dropwise into a PF solution with concentrations of 0.1%, 0.5%, and 1.0% *w*/*v*. The mixture was constantly stirred at a speed of 200 rpm, using a magnetic stirrer (C-MAG HS7, IKA^®^, Staufen, Germany) for a duration of 24 h. Subsequently, the ethanol in PFCU micelle solutions was eliminated using vacuum evaporation (Rotavapor R-124, Büchi Labor Technik AG, Flawil, Switzerland). Then, PFCU was stirred at 200 rpm for 1 h in a water bath at temperatures of 40, 60, and 80 °C, respectively. The PFCU was slowly added into the COS solution (0.1% *w*/*v*) at a ratio of 1:1 while homogenizing. The COSPFCU was centrifuged at 8000× *g* for 15 min at 4 °C (Suprema 21, Tomy, Tokyo, Japan) and then the supernatant was collected. The supernatant was lyophilized by freeze drying (Scanvac Coolsafe Touch 95-15, Labo Gene Aps, Lillerød, Denmark) for 48 h. The condenser temperature and pressure of the freeze dryer were set at −50 °C and 2 mbar, respectively.

The encapsulated CU of samples was determined using a UV-VIS Spectrophotometer (UV-1900i, Shimadzu Corporation, Kyoto, Japan) at the wavelength of 427 nm. The percentage of encapsulation efficiency (%EE) and loading capacity (%LC) were calculated by the following equations:$$Encapsulation\;efficiency\left(\%\right)=\frac{Encapsulated\;curcumin}{Total\;curcumin\;added}\times 100\;Loading\;capacity(\%)=\frac{Encapsulated\;curcumin}{Total\;polymeric\;micelle\;weight}\times 100$$

### 2.3. Characterization of CU-Loaded Polymeric Micelles

The dynamic light scattering (DLS) technique was used for studying hydrodynamic diameter, polydispersity index (PDI), and zeta potential of CU-loaded polymeric micelles (COSPFCU) at 25 ± 0.5 °C by using a Zetasizer (Nano-ZS, Malvern Panalytical Ltd., Malvern, UK).

The morphology of the polymeric micelles was assessed using a transmission electron microscope (JEM-1200, Jeol, Peabody, MA, USA). The COSPFCU sample was subjected to negative staining using a phosphotungstic acid solution on copper grids for a duration of 20 min. Afterward, the surplus phosphotungstic acid solution was removed by employing filter paper. Subsequently, the samples underwent a process of air drying and then were imaged by TEM.

### 2.4. X-ray Diffraction (XRD)

XRD patterns of materials and COSPFCU powder were determined using X-ray diffractometry (D8, Bruker, Billerica, MA, USA) and measured at 40 kV, 40 mA, and 2θ with the scan angle from 5° to 40°.

### 2.5. Spectroscopy Studies

The infrared spectra of COSPFCU and materials were scanned with a Fourier-transform infrared spectroscopy (FTIR) analysis (Spectrum two, PerkinElmer, Waltham, MA, USA) from 4000 to 400 cm^−1^ with a resolution of 4 cm^−1^ at 25 °C. Lyophilized COSPFCU was measured with the KBr.

The proton nuclear magnetic resonance (^1^H NMR) of COSPFCU polymeric micelles was determined (AVANCENEO 400 MHz spectrometer, Bruker, MA, USA) as performed in the ranges of 0.0–11.0 ppm. The COS, PF, and COSPFCU were dissolved by D_2_O, and CU with DMSO-d6.

### 2.6. In Vitro Curcumin Release Assay

The dialysis method was employed to investigate the in vitro release profiles of polymeric micelles, using a previously described method with some modifications [36]. The 2 mL of COSPFCU, which had a concentration of 1.5 mg/mL CU content, was added into a dialysis bag with a molecular weight cut-off (MWCO) of 3500 (CelluSep® CB-501-46, Membrane Filtration Products Inc., Seguin, TX, USA). The release studies were conducted by immersing the bag in a flask containing 250 mL of PBS at pH 7.4: ethanol with a volume ratio of 85:15 (*v*/*v*) which created sink-like conditions. The flask was placed in a shaking incubator (SHKE481HP, Thermo Scientific™, Waltham, MA, USA) and shaken at a speed of 200 rpm at a temperature of 37.5 ± 0.5 °C for a duration of 72 h. Throughout the period from 0 to 72 h, a volume of 2 mL of medium was withdrawn and substituted with fresh medium. Measurement of the CU content was conducted using the method specified in Section 3.2, as previously mentioned. The cumulative percentage release was calculated as:$$Cumulative\;release=\frac{Volume\;of\;sample\;withdrawn\;(mL)}{Bath\;volume\;(mL)}\times P\left(t-1\right)+Pt$$
where Pt = Percentage release at time ‘t’ and P (t − 1) = Percentage release prior to time ‘t’.

### 2.7. DPPH (2,2-Diphenyl-2-Picrylhydrazyl Hydrate) Scavenging Assay

DPPH (2,2-Diphenyl-2-picrylhydrazyl) radical scavenging activity was carried out with some modification [37]. A mixture was prepared by combining 100 µL of a 0.1 mM solution of DPPH with the sample at concentrations ranging from 0.001 to 1 mg/mL. After that, the solutions containing a mixture of both substances were incubated for a duration of 30 min at room temperature under conditions of darkness. The absorbance of the mixes was quantified at a wavelength of 517 nm using a microplate reader (Spectro Star Nano, BMG Labtech, Ortenberg, Germany). The quantification of DPPH radical scavenging activity was conducted by measuring the absorbance values (OD) in each sample to the control, and afterward comparing it to ascorbic acid, which served as the positive control. The DPPH radical scavenging activity was determined using the following calculation method:$$Scavenging\;activity\left(\%\right)=\frac{OD\;control-OD\;sample}{OD\;control}\times 100$$

### 2.8. Cell Line Study

RAW 264.7 macrophage cells were obtained from the American Type Culture Collection. RAW 264.7 cells were cultured in RPMI-1640 medium supplemented with 10% *v*/*v* fetal bovine serum, 1% *v*/*v* double antibody, and 5% CO_2_, at 37 °C constant temperature culture.

#### 2.8.1. Cytotoxicity

The cytotoxicity of CU, COS, and COSPFCU was measured according to the method described previously with slight modifications [38]. RAW 264.7 cells were seeded in 96-well plates (100 μL/well) at 1.5 × 10^4^ cells/well and then cultured overnight. The cells were incubated with COS, CU, and COSPFCU at a concentration between 1.56 µg/mL and 1000 µg/mL in the well, and subsequently were incubated with the cells for 24 h at 37 °C. The blank was added with an equal volume of the complete medium. Then, the cell cultures were added with 10 μL of CCK-8 reagent to each well and incubated at 37 °C for 1–4 h and measured using a microplate reader at a wavelength of 450 nm.

#### 2.8.2. Inhibition of Nitric Oxide Production in LPS-Stimulated RAW 264.7 Cells

The anti-inflammatory activity of CU and COSPFCU was evaluated by NO determination according to the method described previously with slight modifications [39]. RAW 264.7 cells were seeded in 96-well plates (100 μL/well) at 2.0 × 10^4^ cells/well and cultured overnight. Subsequently, RAW 264.7 cells were pre-treated with varying doses of CU and COSPFCU for 12 h. Afterward, the cell lines were subjected to inflammation with lipopolysaccharide (LPS) at a concentration of 50 ng/mL for a duration of 24 h. After the culturing, cell supernatants were collected and tested with a NO detection kit. Then, the solution was measured at an absorbance of 540 nm using a microplate reader. The NO concentration was measured by calculation with a standard curve of sodium nitrite.
$$NO\;inhibition\;rate\;(\%)=\frac{NO\;control-NO\;samples}{NO\;control-NO\;blank}\times 100$$

## 3. Results and Discussion

### 3.1. Fabrication of COSPFCU Polymeric Micelles

The CU-loaded polymeric micelles were fabricated with COS, PF, and CU using the solvent evaporation method [35]. The formation was prompted by a reduction in free energy resulting from elimination of hydrophobic entities within the molecular composition and re-establishment of hydrogen bonds within the aqueous medium. Furthermore, an increase in energy arises due to the Van der Waals interactions between hydrophobic segments within the core of the polymeric micelles [40].

The effects of PF concentration (0.1, 0.5, and 1.0% *w*/*v*), and temperature (40, 60, and 80 OC) on percent encapsulation efficiency (%EE) and percent loading capacity (%LC) of COSPFCU polymeric micelles are demonstrated in Figure 2a,b, respectively. From this result, it is obvious that with an increase in PF content and temperature, polymeric micelles showed an increase in %EE and %LC. However, as the PF concentration rose to 1.0% *w*/*v*, it showed a decline in %EE and %LC. The highest %EE and %LC were achieved at 75.57 ± 2.35% and 10.32 ± 0.59%, respectively, when using 0.5% *w*/*v* PF at 80 °C. It has been reported that the solubility of CU significantly increased (to 21 mg/g) in a self-microemulsifying drug delivery system [41]. Similar to a previous study, the concentration of ferrous glycinate reached a saturation point under optimal conditions [42]. For CU-loaded alginate/chitosan/pluronic F-127 nanoparticles, exceeding the optimal point led to a decrease in the %EE of CU [21].

The solubilization of CU was studied at different processing temperatures of 40, 60, and 80 °C. The amount of CU loaded in COSPFCU polymeric micelles increased remarkably with the increased process temperature, which showed improvement in %EE and %LC of polymeric micelles. For the 0.5% *w*/*v* concentration of PF, the %EE of COSPFCU polymeric micelles increased from 16.53 ± 0.12% to 75.57 ± 2.35% and the %LC increased from 2.36 ± 0.11% to 10.32 ± 0.59%. The high temperatures had a positive effect on CU loading in polymeric micelles by decreasing the critical micelle concentration (CMC) of PF and increasing the solubilization of CU in the formulation [41]. Moreover, the higher temperature in the process caused a disruption of the H-bond between the water molecule and PPO block of the PF wherein the dehydrated PPO blocks aggregated to form the core, while the hydrated PEO blocks constituted the corona [42]. According to previous reports, the amount of lamotrigine increased in nanoparticles with poloxamer when the temperature of the nanoparticles preparation increased from 37 °C to 57 °C [23]. Several reports indicated that COS could improve the water solubility of polyphenol compounds [33,34,43]. Therefore, fabrication of polymeric micelles with a small amount of COS as the outer shell also promoted the CU content solubilizing in formulation better than using the PF alone.

COSPFCU polymeric micelles fabricated with 0.5% *w*/*v* PF at 80 °C were therefore selected to evaluate the characteristics and biological activity in the next section.

### 3.2. Characterization of COSPFCU Polymeric Micelles

The size and morphological characteristics of COSPFCU polymeric micelles were determined by dynamic light scattering (DLS) analysis and transmission electron microscopy (TEM). The COSPFCU micelles had an average size of 73.89 ± 11.69 nm, as well as 230.57 ± 9.17 nm for the particles of COSCU (without PF). For the fabrication of COSPFCU micelles in the aqueous phase, the increased temperature resulted in the formation of closely packed micelles and reduced their hydrodynamic diameter [23]. The polymeric micelles exhibited a positive zeta-potential of 9.89 ± 0.21, which was attributed to the protonation of the −NH_2_ groups of COS to −NH_3_^+^ groups. In addition, the COSPFCU showed a PDI value of 0.285 ± 0.019. Usually, a PDI value below 0.3 is deemed satisfactory for a monodisperse and homogeneous population of nano-formulation for drug delivery systems [37].

The image of COSPFCU polymeric micelles powder is presented in Figure 3a. The morphological image obtained by TEM is shown in Figure 3b, which indicates that the COSPFCU polymeric micelles had a diameter of under 50 nm and were spherical in shape. The hydrodynamic diameter analyzed by DLS was larger than the particle size obtained by TEM due to the dehydrated state of the sample in the analysis [44]. These results confirmed the formation of COS-PF polymeric micelles containing CU in the formulation [45].

### 3.3. X-ray Diffraction Analysis (XRD)

The crystallinity of COS, PF, CU, and COSPFCU was investigated by an X-ray diffractometer, and the XRD patterns are illustrated in Figure 4. The strong peaks of −N-CO-CH_3_ and −NH_2_ of COS are shown in Figure 4a, approximately 11–13° and 22–24°, respectively, according to previous reports [46]. Several characteristic peaks of free CU showed the traits of a high crystalline structure (Figure 4c). The PF and COSPFCU XRD patterns are shown with characteristics of −PEO group peaks at approximately 23° and 28–29° (Figure 4b,d) [44]. The decreased intensity of two major peaks of PF in COSPFCU confirmed the formation of polymeric micelles. In COSPFCU, the crystalline structures being in an amorphous state caused a significant decrease in intramolecular H-bonds before the conformation [47,48]. However, the XRD pattern of COSPFCU did not show the fingerprint of COS and CU due to the high concentration of PF and decrease of H-bonds in formation. The observed reduction in crystallinity in the COSPFCU pattern provided evidence supporting the enhanced water solubility of the drug delivery system derived from COS-PF [45].

### 3.4. Spectroscopy Analyses

The chemical structure of COSPFCU was measured by ^1^H NMR and FTIR spectroscopy.

FTIR spectra of COS, PF, CU, and COSPFCU are shown in Figure 5. In the COS spectrum, the bands located at 3540 to 2847 cm^−1^ are attributed to −OH and N-H groups, respectively [49]. The bands in the region of 1630, 1517, and 1320 cm^−1^ are referred to as C=O, amide I, and amide II, respectively [45]. The band at 1381 cm^−1^ is attributed to the vibrations of -CO in the glycosidic ring. The region of 1067 cm^−1^ is assigned to the C-O stretching.

The PF spectrum had peaks at 1100 and 961 cm^−1^ due to the C-O groups. The characteristic peak at 1284 cm^−1^ was assigned to the −CH_2_ groups in the PF, according to previous research [50]. In addition, the PF presented peaks at 3513 and 2885 cm^−1^ which indicated O-H stretching in PEO blocks and C-H from the methylene groups [49].

Within the CU spectrum, there was one peak of the O-H group at 3510 cm^−1^. The characteristic peaks of C=C and C=O correspond to 1626 and 1601 cm^−1^ on the spectrum. The bands in the region of 1508 and 1279 cm^−1^ are referred to as C=O vibration and C-O band from the enol groups, respectively [49]. The characteristic peak of C-H was determined to be at 961 cm^−1^ [50,51].

The FTIR spectrum of COSPFCU illustrated the major peaks of PF but also contained the signals of COS and CU in the spectrum. The spectrum of COSPFCU indicated the O-H vibration at 3399 cm^−1^ and C-H at 2890 cm^−1^. In addition, the peaks around 1342 and 1114 cm^−1^ were assigned to the N-H of COS and C-O of PF, respectively. There was some degree of shifts for some characteristic bands between 1650 and 800 cm^−1^. These results suggested the construction of polymeric micelles.

The ^1^H NMR is illustrated in Figure 6. The typical signals for PF were surrounded at 1.18 and 3.71 ppm, indicating the non-hydroxyl proximal methylene and O-methylene groups, respectively. Within the CU spectrum, the peak at 3.83 refers to the methyl of methoxyl groups. The characteristic peaks of the benzene rings and 1,6-heptadiene-3,5-dione group are shown at 6.61–7.58 ppm, and 9.68 ppm is assigned to the polyphenol hydroxyl groups. The ^1^H NMR spectrum of COS presented the signals of hydrogens on the sugar ring at 3.00–4.20 ppm.

The ^1^H NMR spectrum of the COSPFCU showed the signals of the sugar ring of COS from 3.00 to 4.20 ppm. The peak at 1.18 ppm indicated non-hydroxyl proximal methylene of the PF. The peak at 3.71 ppm in COSPFCU was attributed to O-methylene of PF. These results proved the COS-PF formation in the polymeric micelles with non-covalent bonds. However, weak signals of CU in the COSPFCU were indicated in the range of 6.50–7.50 ppm due to the low concentration of CU in the polymeric micelles (approximately 100 µg/mg of COSPFCU), according to the previous report [52].

The FTIR and ^1^H NMR results indicated the successful formation of COSPFCU polymeric micelles.

### 3.5. In Vitro Release Assay

Drug release studies were conducted for free-CU and COSPFCU using the dialysis method at pH 7.4, as illustrated in Figure 7. Approximately 95% of CU from the CU solution in the dialysis bag was released within 48 h, with an initial rapid release in the first 3 h, followed by continuous release throughout the duration of 6 h. The COSPFCU had released less than 20% of the CU after 3 h and less than 25% after 6 h. After 72 h, 30% of the CU in COSPFCU had been released. Overall, these findings are in accordance with findings reporting that approximately 70% of the CU loaded with pluronic F-68 was released in 72 h [50]. The polymeric micelles with COS and PF delayed the release of the CU from the formulation due to the internal protective force the COSPFCU polymeric micelles containing COS and PF created during the fabrication process when the temperature rose [53].

The result of this study indicated that the CU molecules were effectively encapsulated in the polymeric micelles with the COS and PF as carriers. This kind of polymeric micelles could release drugs in a controlled manner to provide sustained delivery.

### 3.6. DPPH Scavenging Activity of COSPFCU

The antioxidant activities of CU-loaded polymeric micelles and materials are shown in Figure 8a. The increasing concentrations resulted in an improvement in the DPPH scavenging activity of the samples. The DPPH scavenging activity of COS was 30.16% at 1 mg/mL which was related to −OH and −NH_2_ [46]. The COSPFCU showed high antioxidant activity with scavenging activity of 80.49% at 0.5 mg/mL (0.05 mg/mL of CU, 10% *w*/*w* in COSPFCU), comparable to 0.05 mg/mL of CU in ethanol with scavenging activity of 88.73%. When the concentration of COSPFCU increased to 1.0 mg/mL, the inhibition activity did not markedly increase. The antioxidant activity of CU and COSPFCU was mostly endowed by functional groups such as keto-enol groups, aromatic rings, and phenols [54]. Due to the fact that polymeric micelles contained only 10% (*w*/*w*) CU, COSPFCU had an IC50 on DPPH scavenging activity of 0.185 ± 0.0076 mg/mL, which was higher than CU (Figure 8b).

It is notable that CU loaded in polymeric micelles (COSPFCU) showed strong antioxidant activity comparable to the native CU in ethanol, indicating the promise of the water-soluble COSPF as a reliable drug delivery vehicle.

### 3.7. Cytotoxicity Test

As illustrated in Figure 9, the cell viability was evaluated for RAW 264.7 cells treated with COSPFCU polymeric micelles and materials by using the colorimetric CCK-8 kit. The RAW 264.7 cells were treated with samples in different concentration in the range of 1.56–1000 µg/mL for 24 h. CU showed cytotoxicity to RAW 264.7 cells with concentrations higher than 3.125 µg/mL, which was similar to previous reports [38,55]. The COSPFCU polymeric micelles were able to reduce the cytotoxicity significantly when the concentration was below 62.5 µg/mL (CU 6.25 µg/mL, 10% *w*/*w* in COSPFCU). Furthermore, the COS did not show cytotoxicity to the cells which can reduce drug toxicity in nano-formulation to the normal cell lines [56,57].

### 3.8. Anti-Inflammatory Regulation by Inhibition of NO Production

Nitric oxide (NO) is a signaling molecule that plays a pivotal role in the pathophysiology of inflammation and exhibits anti-inflammatory properties under normal conditions. On the other hand, excessive production in atypical circumstances leads to inflammation. NO is involved in the pathogenesis of inflammatory disorders in the joints, intestines, and lungs. Hence, inhibitors of NO production serve as a principal means of regulating inflammatory disorders [58].

LPS stimulation induced a significant increase of NO production in the macrophage cells. The LPS-stimulated RAW 264.7 cells were treated with free CU and COSPFCU, respectively, as illustrated in Figure 10.

Inhibition increased with the increasing concentration. The inhibition of NO activity was 18.46–33.20% at a concentration of 0.78–3.125 µg/mL for CU and 18.00–88.21% for COSPFCU at a concentration of 7.8–125 µg/mL, respectively. But COSPFCU had high cytotoxicity when the concentration was higher than 62.5 µg/mL (Figure 9). The anti-inflammatory properties of CU were attributed to its interaction with receptors and signaling pathways, hence regulating the response of target tissues to inflammatory mediators. Additionally, CU could stimulate the production of anti-inflammatory mediators [59].

The results suggest that the water-soluble COSPFCU had great anti-inflammatory effect by decreasing the production of NO in RAW 264.7 cells. Notably, the anti-inflammatory activity of COSPFCU was much better than CU. Therefore, the COSPFCU with controlled release and high water solubility can reduce the cytotoxicity and improve anti-inflammatory effects.

Finally, all the results demonstrated the potential of COS and PF to promote the water solubility of hydrophobic molecules as well as to effectively control their release.

In summary, the polymeric micelles fabricated from COS and PF showed a high enhancement in water solubility of CU, approximately 9093-fold, compared with native CU 11 ng/mL [15]. And it was also better than COS alone to encapsulate hydrophobic molecules with 3.5-fold [43] and 5.83-fold [34], respectively. Moreover, the increased fabrication temperature led to a decrease in the critical micelle concentration of PF which resulted in improvement in the %EE and %LC of the polymeric micelles. The COSPFCU showed PDI below 0.3 and hydrodynamic diameter as 73.89 ± 11.69 nm. The potential of COSPFCU was evaluated for antioxidant activity, cytotoxicity, and anti-inflammatory activity. The COSPFCU in water showed good DPPH scavenging ability of 80.49 ± 0.51% at 0.5 mg/mL (0.05 mg/mL of CU, 10% *w*/*w* in COSPFCU) comparable to 88.73 ± 1.85 of CU at 0.05 mg/mL in ethanol. Furthermore, the polymeric micelles with COS and PF could reduce the cytotoxicity of CU to the RAW 264.7 cells and improve the inhibition of NO production. These results confirmed the potential of the COS-PF polymeric micelles system to encapsulate hydrophobic molecules to improve the characteristics and biological activities of the drug.

## 4. Conclusions

The CU-loaded polymeric micelles fabricated with COS and PF demonstrated enhanced solubility, controlled release, and better biological activity. The polymeric micelles (COSPFCU) composed of COS and PF illustrated the highest encapsulation efficiency and loading capacity at the PF’s concentration of 0.5% *w*/*v* and fabrication temperature of 80 °C. The COSPFCU showed high antioxidant activity and anti-inflammatory activity, as well as reduced cytotoxicity in the RAW 264.7 cells. In addition, the polymeric micelles could release CU in a controlled manner and provide sustained delivery. For these reasons, the potential of the nano-formulation from the COS and PF was confirmed by the polymeric micelle methods, which can be a candidate for encapsulation of the hydrophobic molecules and application in pharmaceutical and medical fields.

## Figures and Tables

**Figure 1 pharmaceutics-15-02497-f001:**
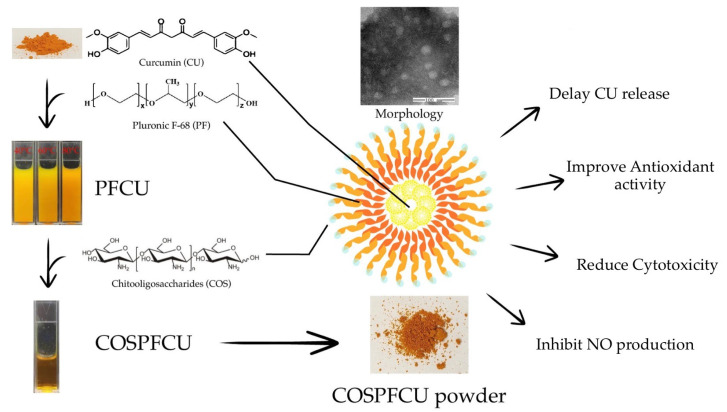
The scheme of preparation and primary bioactivity evaluation of novel water-soluble curcumin-loaded polymeric micelles (COSPFCU) fabricated with chitooligosaccharides (COS) and pluronic F-68 (PF).

**Figure 2 pharmaceutics-15-02497-f002:**
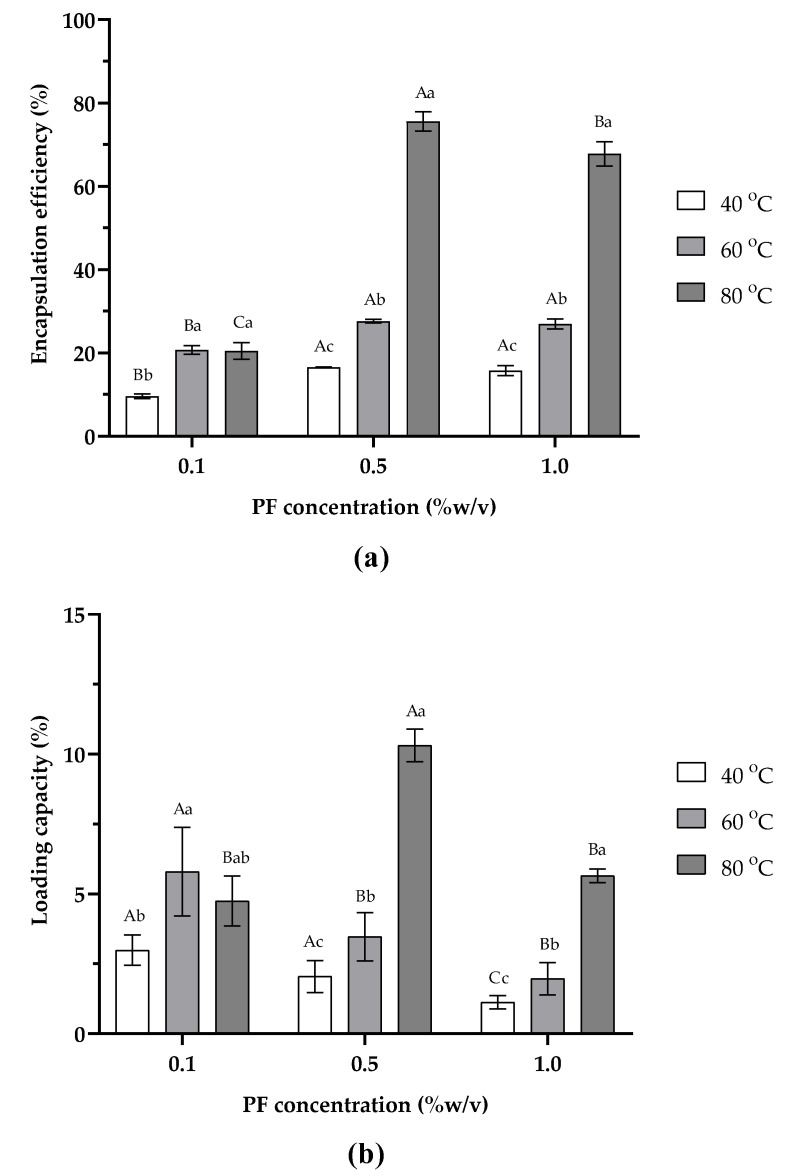
Effects of pluronic F-68 (PF) concentration and fabrication temperature on polymeric micelle characteristics: (**a**) encapsulation efficiency, (**b**) loading capacity. Values with capital letters indicate significant differences between concentration of PF at the same process temperature (*p* < 0.05). Different small letters indicate significant differences within process temperature in the same concentration (*p* < 0.05).

**Figure 3 pharmaceutics-15-02497-f003:**
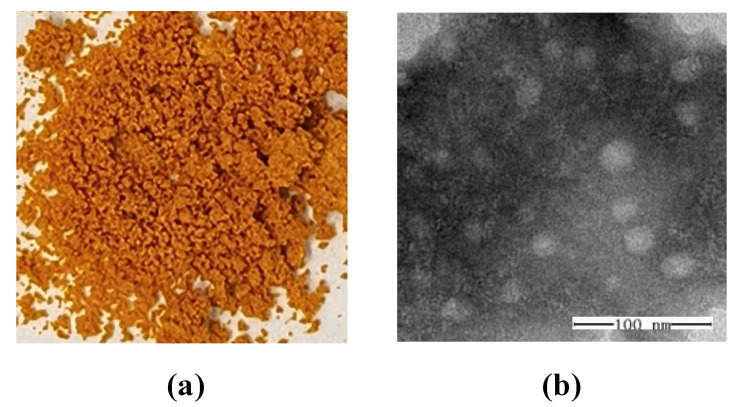
(**a**) The appearance of curcumin-loaded polymeric micelles (COSPFCU). (**b**) The morphology of COSPFCU using transmission electron microscopy at 300,000× magnification.

**Figure 4 pharmaceutics-15-02497-f004:**
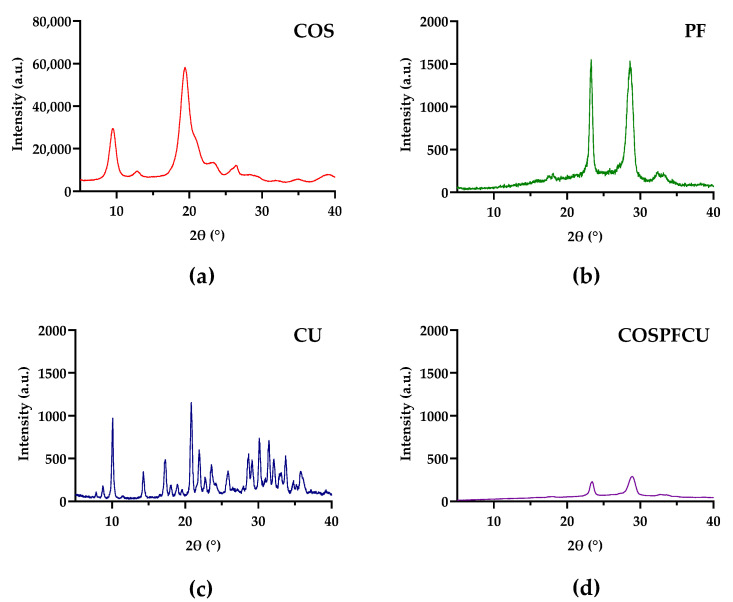
The XRD patterns of (**a**) chitooligosaccharides (COS); (**b**) pluronic-F68 (PF); (**c**) curcumin (CU); (**d**) CU-loaded polymeric micelles (COSPFCU).

**Figure 5 pharmaceutics-15-02497-f005:**
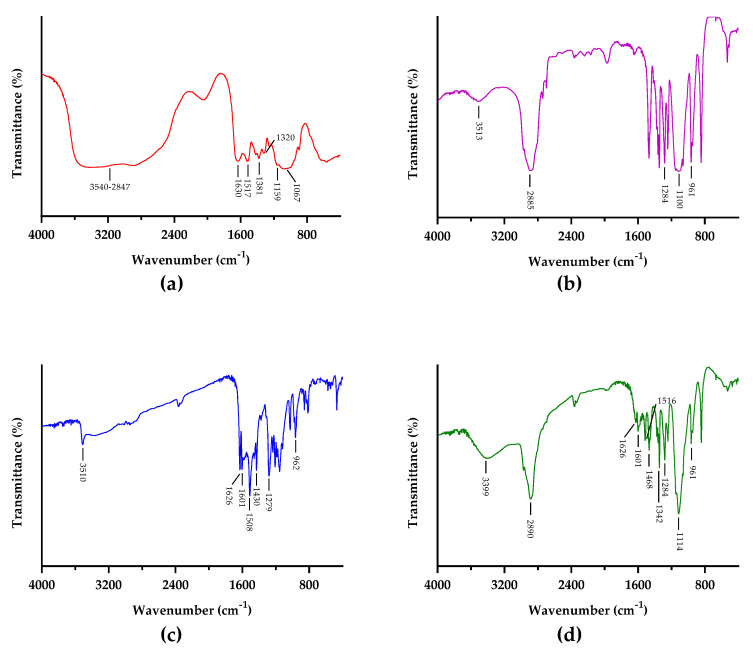
FTIR spectra of (**a**) chitooligosaccharides (COS); (**b**) pluronic-F68 (PF); (**c**) curcumin (CU); (**d**) CU-loaded polymeric micelles (COSPFCU).

**Figure 6 pharmaceutics-15-02497-f006:**
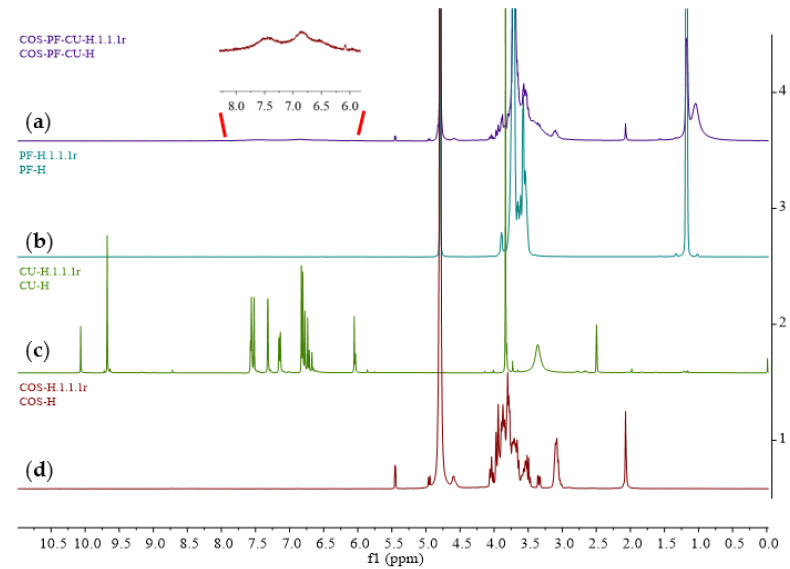
The ^1^H NMR spectra of (**a**) curcumin-loaded polymeric micelles (COSPFCU); (**b**) pluronic F-68 (PF); (**c**) curcumin (CU); (**d**) chitooligosaccharides (COS).

**Figure 7 pharmaceutics-15-02497-f007:**
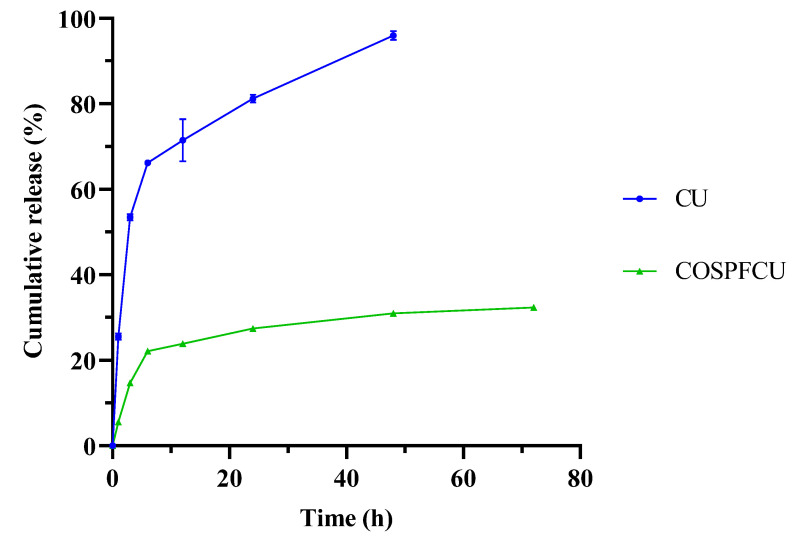
In vitro release of free-curcumin (CU) and CU-loaded polymeric micelles (COSPFCU) in phosphate buffer saline at pH 7.4.

**Figure 8 pharmaceutics-15-02497-f008:**
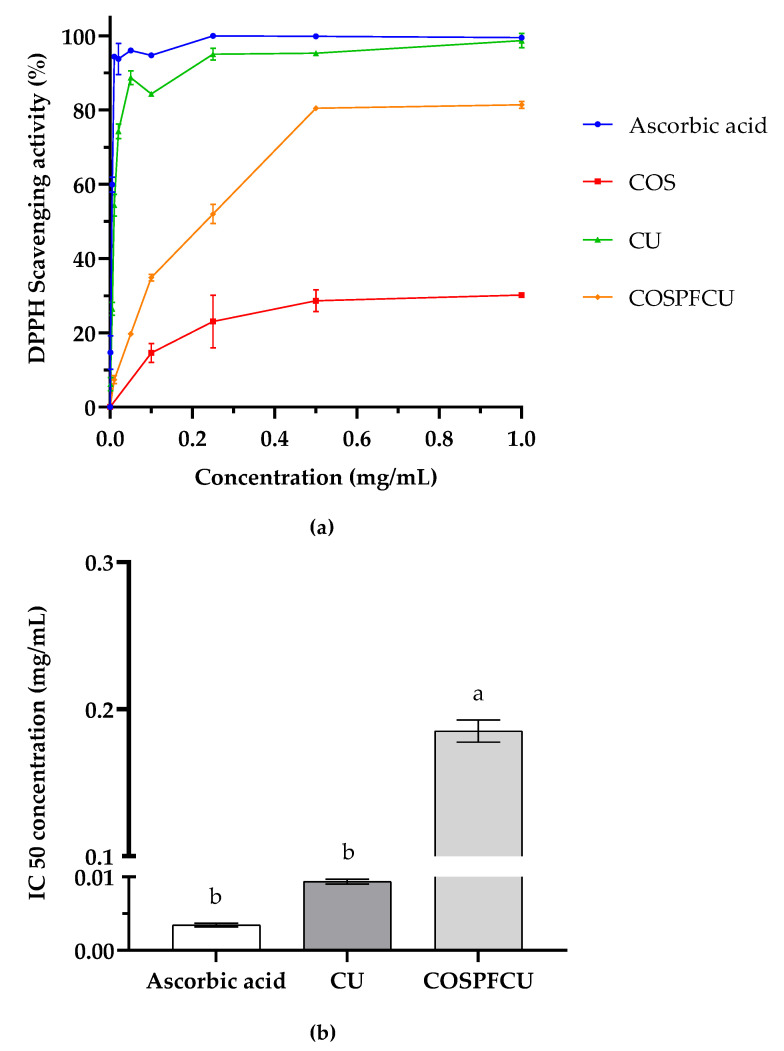
(**a**) DPPH scavenging activity of chitooligosaccharides (COS), curcumin (CU), and CU-loaded polymeric micelles (COSPFCU) with ascorbic acid used as the positive control. (**b**) The half maximal inhibitory concentration of DPPH scavenging activity (IC50). Different small letters indicate significant differences within sample (*p* < 0.05).

**Figure 9 pharmaceutics-15-02497-f009:**
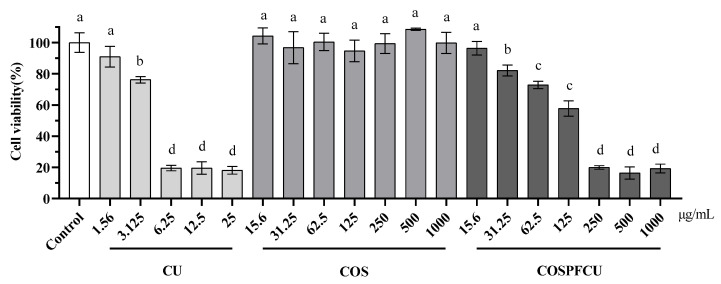
The cell viability of RAW 264.7 cells after treatment with curcumin (CU), chitooligosaccharides (COS), and CU-loaded polymeric micelle (COSPFCU) at various concentrations following 24 h. Different small letters indicate significant differences within sample and concentration (*p* < 0.05).

**Figure 10 pharmaceutics-15-02497-f010:**
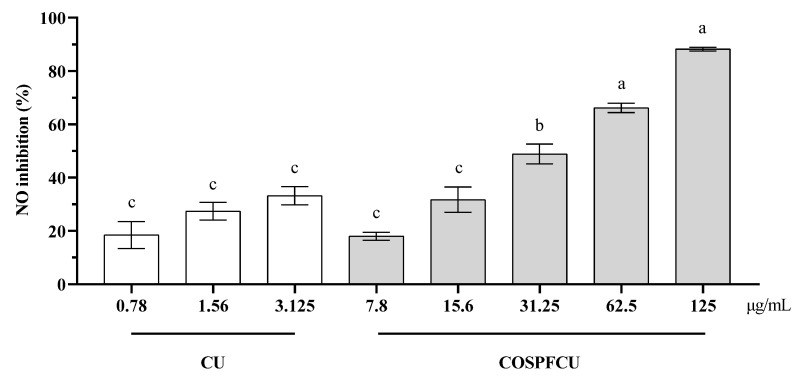
The inhibition of NO in RAW 264.7 cells after LPS co-treatment with curcumin (CU), CU-loaded polymeric micelles (COSPFCU) at various concentrations following 24 h. Different small letters indicate significant differences within samples and concentrations (*p* < 0.05).

## Data Availability

Not applicable.

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
