# Peer review of "Preparation and Primary Bioactivity Evaluation of Novel Water-Soluble Curcumin-Loaded Polymeric Micelles Fabricated with Chitooligosaccharides and Pluronic F-68"

_pharmaceutics, 2023, doi:10.3390/pharmaceutics15102497_

Round 1
Reviewer 1 Report
Li and co-workers report on the preparation and biological evaluation of curcumin nanocarriers based on pluronic F-68 and an oligosaccharide. The novel nanocarriers show high curcumin encapsulation and interesting bioactivity. The work shows originality but some points should be further clarified.
1. The role of oligosaccharide in the structure of the nanocarriers needs to be further researched and elucidated. What are the interactions that bind the two nanocarrier components (F-68 and COS)? What is the location of COS in the nanocarriers? Location at the nanocarrier corona is assumed but some further experimental proof, by NMR or FTIR, should be provided. Are the F-68/COS mixed aggregates still of micellar nature?
2. What is the reason for better solubilization of CU in the micelles as temperature increases? The PPO component is expected to become more hydrophobic at higher temperatures. Are there any structural changes of the mixed F-68/COS micelles with temperature? Further studies are needed on empty F-68/COS micelles.
3. Line 175: how do FTIR measurements indicate successful fabrication of micelles? The technique merely indicates the presence of the components in the mixed carriers. Please discuss further.
4. XRD studies should be better connected to the solution structure of mixed micelles.
5. Section 3.8.2: change heading to be more descriptive.
6. Use of English should be improved substantially. Several grammar and syntax errors should be corrected.
Use of English should be improved substantially. Several grammar and syntax errors should be corrected.
Author Response
To, Reviewer 1
The manuscript by Ingrungruengluet et al. demonstrated research evidence about “Preparation and Primary Bioactivity Evaluation of Novel Water-Soluble Curcumin-loaded Polymeric Micelles fabricated with Chitooligosaccharides and Pluronic F-68”. The manuscript represented interesting data; however, some revisions are needed.
Firstly, we would like to express our gratitude for your thoughtful comments and suggestions. Your expertise and attention to details have been instrumental in improving the article, and we are sincerely appreciative of your contribution. Your review has made a critical and thoughtful impact on our work. We have carefully considered your feedback and made the necessary revisions based on your suggestions.
Sincerely,
Pattarachat Ingrungruengluet

Reviewer 2 Report
The paper submitted by Ingrungruengluet et al. investigated the preparation of Curcumin-loaded Pluronic micelles coated with oligochitosan. Even if the idea of the paper is not very original, the study is interesting. However, some modifications are needed:
1. the micellization of different types of Pluronics are very well studied and the other must complete the introduction section. A recommendation can be: https://doi.org/10.3390/polym14153007
2. fig 3: from fig 3b it seems that the size of micelles are smaller than 50 nm.
3. what are the micellar sizes before the addition of COS?
4. section 2.4: the authors must discuss about the shift of the characteristic peaks of COS, PF and CU in the COSPFCU. These shifts are indicative of the interactions between the polymers and drug and can be of importance for the drug release. A model of discussion can be found here: https://doi.org/10.3390/polym12071450
5. line 191: which is the meaning of "CU molecules were well-encapsulated"? From fig 6 it seems that there is a burst effect in the first 8h which means that CU was also adsorbed at the surface of the micelles.
Language is quite poor and the manuscript must be checked by an native English speaker.
Author Response
To, Reviewer 2
The manuscript by Ingrungruengluet et al. demonstrated research evidence about “Preparation and Primary Bioactivity Evaluation of Novel Water-Soluble Curcumin-loaded Polymeric Micelles fabricated with Chitooligosaccharides and Pluronic F-68”. The manuscript represented interesting data; however, some revisions are needed.
Firstly, we would like to express our gratitude for your thoughtful comments and suggestions. Your expertise and attention to details have been instrumental in improving the article, and we are sincerely appreciative of your contribution. Your review has made a critical and thoughtful impact on our work. We have carefully considered your feedback and made the necessary revisions based on your suggestions.
Sincerely,
Pattarachat Ingrungruengluet

Round 2
Reviewer 1 Report
The authors have revised the manuscript taking into account all my comments.
The manuscript has been improved substantially. Therefore, I recommend its publication in the current form.
Reviewer 2 Report
The paper can be accepted as it is.
No issues detected.